# Features of Engorgement of *Ixodes ricinus* Ticks Infesting the Northern White-Breasted Hedgehog in an Urban Park

**DOI:** 10.3390/microorganisms11040881

**Published:** 2023-03-29

**Authors:** László Egyed, Dávidné Nagy, Zsolt Lang

**Affiliations:** 1Veterinary Medical Research Institute, 1143 Budapest, Hungary; 2Healthware Consulting Ltd., 1093 Budapest, Hungary; davidne.nagy@gmail.com; 3Department of Biomathematics and Informatics, University of Veterinary Medicine, 1078 Budapest, Hungary; lang.zsolt@univet.hu

**Keywords:** *Ixodes ricinus*, hedgehogs, urban park, sampled engorged ticks, seasonality, engorgement time

## Abstract

In this work we exploited the parallel dense tick and hedgehog populations of an urban park in Budapest, Hungary as a good host–parasite model to obtain detailed data about this physiological relationship. Over a 27-week period from April to October, 57 hedgehogs were captured in an urban park and kept for 10–14 days in animal house. All dropped off ticks were sampled, which allowed us to draw more a detailed picture of *Ixodes ricinus*–hedgehog relationships. The results indicated that the hedgehog is an effective host for ticks (prevalence: 100%) and the mean intensity of infestation was 83.25. Of the male ticks, 68.42% dropped off dead; 1.56% of the dropped off nymphs and 11.4% of the larvae finished their bloodmeal with red cuticles, while 5.79% of the females could not finish their blood meal, and dropped off dried, dead, or shrunken. We applied novel statistical methods of survival analysis of prevalent cohorts to estimate the whole attachment times of ticks from the observed attachment times, having no information about when the ticks attached to their hosts. Mean attachment times were 4 days for larvae, 5 days for nymphs, 10 days for females, and 8 days for males. On the first day after capture of the hosts, fewer females, nymphs, and larvae detached engorged than had been predicted, but this was not true for males. Mean intensity of infestation per host was 1.4 for males, 6.7 for females, 45.0 for nymphs, and 29.3 for larvae. As regards seasonality, the activity of all stages of ticks consisted of several smaller peaks and considerably differed by season. Studies of the dense tick–host populations of this natural habitat could provide further valuable data about tick–host relations, the data of which cannot be drawn from most other hedgehog habitats.

## 1. Introduction

Hedgehogs are frequent natural hosts for all life stages of ticks, move slowly, with their abdominal surface close to the ground, they are active at sunset and sunrise, which are also the main activity periods for ticks [1], their grooming activity is not efficient (short legs, rigid neck), and they dig burrows where they sleep motionless for hours during daytime. Their habitats are mainly forests, and the marginal zones of forests, and shady, abandoned bushy areas are environments where ticks also tend to accumulate. In Europe, east of Italy, Austria, and Germany extending to Siberia, the northern, white-breasted hedgehog (*Erinaceus roumanicus*) lives and is present also in Hungary.

The most common tick species parasitizing the European hedgehog are *Ixodes ricinus* and *Ixodes hexagonus* [2]. Prevalence and seasonal fluctuations of the proportion of *Ixodes hexagonus* (a nest-dwelling hedgehog specialist species) are much lower than in the generalist *I. ricinus* [3]. During a study at a Danube island (which is our study site) 1.12% *I. hexagonus* were found among sampled ticks [4].

By studying an urban habitat with parallel dense *E. roumanicus* and *I. ricinus* populations, our aim was to analyse various features of the host/parasite relationship, seasonality, and the questing and feeding activities of hedgehog-infesting ticks.

## 2. Materials and Methods

### 2.1. Study Design

The study site was on Margaret Island (0.965 km^2^), which is situated on the Danube, in the centre of Budapest, the capital of Hungary. It is a regularly managed, watered public park, with hundreds of trees, trimmed grasslands, planted flowers, and large areas covered by dense layer of common ivy (*Hedera helix*). The Danube probably increases the level of air humidity and moderates maximal and minimal daily temperatures. The local hedgehog population is not under notable pressure from carnivorous animals, although dogs of visitors might disturb them.

The time of the study covered the activity period of European hedgehogs, when the mean daily temperature exceeds 8 °C [5]. The first hedgehog was captured on 1 April, while the last on 16 October, which almost coincided with the activity period of *I. ricinus*. Hedgehogs were captured on Mondays exactly after sunset, (two individuals on each occasion) always on the same 25-hectare part of the island. Three times only one hedgehog was found, and in these cases, sampling was repeated next day for the second animal. Three times three hedgehogs were captured on one day.

The captured animals were kept in the animal house of Veterinary Medical Research Institute, in rabbit cages, fed on dry cat food and water, for a minimum of 8 days, or until no engorged ticks dropped off them for two consecutive days. The study was permitted and reviewed by the ethical committee of the institute (permission number: PE-06/KTF/21399-7/2022). In the morning after their capture, the animals were examined for the sites of feeding tick females, their body weight was measured, and their sex was determined. The detached engorged ticks were sampled and counted daily from an aluminium tray put below the cages. To avoid possible immediate recaptures, hedgehogs were always captured on Mondays and were set free on Tuesdays next week (if on the last two days no detached engorged ticks were observed).

Maximal, minimal, and mean daily temperatures and precipitation data were collected from the homepage of the National Meteorological Service (http://www.metnet.hu), accessed on 1 March–15 October 2022.

### 2.2. Statistical Analysis

All the statistical analyses and computations were carried out by the use of the R 4.0.4 statistical software [6].

The observed attachment time of a tick lasted from the capture of its host until the tick dropped off. It was set to 0.5 days for ticks that fell off during the first day following the capture of their hosts, 1.5 days for ticks that fell off on the second day, etc. This duration is typically much shorter than the whole attachment time of the tick measured from the day the tick lands on the hedgehog until it falls off. When a hedgehog is captured, the ticks residing on it form a so-called prevalent cohort. In the statistical literature of prevalent cohorts, the observed attachment time of a tick is called ‘forward recurrence time’. The time elapsed from the onset of attachment of a tick until the capture of its host is called ‘backward recurrence time’ [7]. The methods [8] used for estimating the distribution of the whole follow-up time are based on observed backward or forward recurrence times for prevalent cohort data. The conditions of applicability of these methods are as follows: the date of capture is supposed to be independent of the magnitude of the whole attachment time, and uniformly distributed over the whole attachment time of any tick that belongs to the prevalent cohort. Moreover, it is assumed that the probability for ticks to be sampled is proportional to the length of their whole attachment time. If these conditions are met, then the distribution of the observed attachment time can be transformed into the distribution of the whole attachment time and vice versa. Mathematical details were presented [9].

Weibull distribution was adapted to the whole attachment times. This is a flexible distribution family that depends on a scale and a shape parameter, and it is frequently selected in parametric models of time to event data [10]. The corresponding distribution of the observed attachment times belongs to the family of generalised gamma distributions [11]. It fitted our observed attachment time data reasonably well. Formulas are referred to in Appendix A.

We applied an accelerated failure time (AFT) model [10] to the observed attachment times (i.e., the forward recurrence times). Briefly, AFT is a regression model for log transformed time to event outcome variables. The explanatory variables were tick sexes stages, season (spring, summer, autumn), and their interaction. Another AFT model was demonstrated [12] relating the whole follow-up times (i.e., the whole attachment times of the ticks) to the same explanatory variables having the same coefficients [8]. Applying the method in [12], we carried out statistical inference for the whole attachment times based on the results obtained from the AFT model fitted to the observed attachment times.

The observed data of ticks harboured by hedgehogs were clustered and correlated, because ticks attached to the same hedgehog host had shared exposures and characteristics. Therefore, to obtain approximately unbiased estimates of the parameters and their covariance matrix, we applied a marginal AFT model [13] (i.e., a GEE model) using the survreg procedure of the R 4.0.4 package survival [14]. The cluster parameter of the survreg procedure was selected to be the identifier of hedgehogs in the collected data set. We noted that the current version of survreg requires that the value of the shape parameter of the distribution of the observed attachment times (see Equations (A1) and (A2) in Appendix A) be given in advance, thereby this parameter and its standard error cannot be estimated by survreg immediately. Technical details of how we overcame this limitation are relegated to Appendix B. 

The ratio of the expected mean of the whole attachment time to the expected mean of the observed attachment time depends only on the shape parameter. The formula is Equation (A4) in Appendix C. From (A4) an estimate and a 95% confidence interval for this ratio was obtained based on the estimated shape parameter and its standard error.

We observed that on the first day following the capture of hedgehogs, fewer ticks fell off the hosts than on the second day. According to our statistical model, observed attachment times had a monotone decreasing density function. See Equation (A2) in Appendix A. Consequently, the observed small frequencies of ticks dropped off on the first day following capture may have violated somewhat the assumptions of our model. To test the difference of the fall-off frequencies between the first and second day statistically, we fitted a negative binomial generalised linear mixed model to the number of ticks that dropped off on the first and second days, respectively. Tick life stage was included in the model as an explanatory variable. To account for the clustered nature of the data the hedgehog hosts were also included as random subjects in the model. Procedure glmmTMB from the R 4.0.4 package glmmTMB [15] was applied to fit the model and carry out the test. To investigate the magnitude of effects related to the potential violation of assumptions we carried out a sensitivity analysis on our model by assigning the common observed attachment time of 1 day to all ticks that fell off on the first two days following the captures of their hosts.

We also analysed the abundances of ticks harboured by hedgehog hosts. A marginal generalised linear model, i.e., a GEE model, [13] fitting Poisson distribution was applied to tick abundance data. This model accounted for the correlation between abundances of ticks belonging to different stages and attached to the same hedgehog. The explanatory variables were tick life stage, season (spring, summer, autumn), and their interaction. Procedure geeglm from the R 4.0.4 package geepack [16] was used to fit the model.

To compare the modelled means of the whole attachment times and modelled means of the abundances of ticks harboured by hedgehog hosts between tick life stages within each season and also between seasons within each tick life stage, we applied *post hoc* multiple comparison tests [17]. The R 4.0.4 procedures needed were emmobj from R 4.0.4 package emmeans [18] and glht from package multcomp [17].

Throughout this study the level of significance was set to 5%.

## 3. Results and Discussion

### 3.1. Collected Hedgehogs and Ticks

Under the effects of the Danube and the artificial urban environment, abundant hedgehog and tick populations could develop in parallel in our sampling territory. In contrast to this island, in natural sites (forests), hedgehog populations are much less dense. The special ecological situation of the Margaret Island obviously provides optimal conditions for studying the hedgehog–tick relations.

All of the hedgehogs were parasitised by ticks with an average intensity of 83.25 (varied between 4 and 300). The prevalence values were 59.65% for males, 84.21% for females, 96.5% for nymphs, and 100% for larvae.

Over the 27-week sampling period, a total of 57 hedgehogs were captured, 32 (56%) were females, and 25 (44%) were males. Eight juvenile hedgehogs (body weight 0.24–0.44 kg) including three males and five females were also caught. The average body weight of hedgehogs was 0.65 kg (adults 0.72 kg, juveniles 0.35 kg). All animals were active and healthy. A total of 4745 ticks fell off the captured hosts, 99.35% of them were *I. ricinus* (75 males, 375 females, 2607 nymphs, 1657 larvae), while 31 (0.65%) *Ixodes hexagonus* nymphs were found. For detailed seasonality data see Table 1.

On the abdominal site of 16 hosts, feeding females were observed. Most (90%) of the observed engorging females were seen on the thoracic and axillary regions, and the remaining few on the abdominal area. Ticks were never seen on the hind legs, head, auricles, and the spiny dorsal part of the body. As we could not find all females and no nymphs and larvae these data could not be statistically analysed.

### 3.2. Abundance of Ticks

The highest abundance of both sexes of questing and feeding ticks was recorded in spring. The number of males sharply and that of females slowly declined over the following seasons (50–14–15 males, and 216–103–61 females in spring, summer, and autumn, respectively). Nymphs were most abundant in spring. Later their abundance decreased slightly (by 23.7%) in summer, and they almost disappeared by autumn (1552–1184–148). In spring and summer 57.6% of all ticks found on the hedgehogs were nymphs. Summer was the main season for larvae, but their abundance (304–863–181) was not so high as of nymphs.

The modelled and observed mean tick abundances per hosts are shown in Table 2. The ratios of mean abundances per hosts, their 95% confidence interval values, and *p*-values indicating significant differences between seasons and tick life stages are summarised in Table 3.

### 3.3. Time of Engorgement

Engorged larvae and nymphs dropped off the host during the first 6 and 9 days, respectively. Engorgement of females took 12 days, 90% of the larvae finished blood meal in 5 days, nymphs in 6 days, and females in 8 days. Males left the host continuously, more consistently, and uniformly for 11 days but most of them in 8 days (Figure 1, Table 4).

The attachment times of the tick life stages showed a specific course throughout the year (Figure 2), sometimes with significant differences between the seasons (Table 3). Adults spent continuously shorter times on the host as the year went by, males were present on their hosts for a significantly shorter time in autumn than in summer (*p* = 0.020). Only nymphs had a relatively stable course of their attachment times over the seasons, with least time on the host in summer similar to attachment times of larvae (see Figure 2 and Table 3).

Post hoc tests of mean attachment times of the developmental stages according to seasons revealed that in the spring females spent significantly more time on the host than the males, but this difference was not true for summer and autumn. Nymphs did not spend more time on the host in spring than larvae, but significantly more in the summer and slightly more in the autumn. The times males spent on the hosts were significantly longer than those of the subadult stages throughout the year (Table 3).

The mean attachment times in our statistical model (that lasted from the date when the ticks landed on the hedgehogs until they detached) were 1.88 times longer (95% CI: 1.80–1.91) than the mean observed attachment times (that lasted from the capture of hedgehogs until the ticks dropped off). This was true for all groups (sexes, stages, seasons) studied in the model.

The attachment times of ticks showed no significant relationship to the body weight and sex of the host individuals.

We did not know the exact time when ticks infested their hosts, which made it difficult to evaluate the observed feeding times. Our estimated engorgement time data roughly corresponded to the engorgement times reported in the literature (larvae 2–4, nymphs 3–5, females 6–10 days, [19]; larvae 2–6 days, nymphs 3–7, females 5–14 days, [20]). The modelled times seemed to be slightly longer, while the observed times were shorter.

### 3.4. Ticks Detached on the First Day

During the first day after capture of the hosts, the number of the dropped off engorged ticks was lower than predicted by the statistical model. This decrease was most pronounced in case of nymphs and larvae, less marked in the case of females but was not significant in the case of males. Differences between the numbers of ticks detached on the first and second days after capture of the hedgehogs (Table 4) were statistically significant both for females (*p* = 0.004) and for subadults (*p* < 0.001). Although fewer males dropped off on the first day after capture than on the second and the difference between the values of days 1 and 2 was not significant (*p* = 0.352) (Figure 1, Table 4). 

It is obvious that on the first day after the capture of hedgehogs fewer ticks detached from the hosts than on the subsequent days. This temporary decrease was statistically significant only for the obligatory hematophagous ticks (larvae, nymphs, females), but not for the facultative hematophagous males (Figure 1). We suppose that some factor in the blood or body fluids of the hosts caused this moderate prolongation of the engorgement process. As the capture of the hosts is a stress factor, we hypothesise that the temporarily elevated stress hormones (glucocorticoids) could have caused such an effect. From the second day after the capture detachment of ticks went on, as the statistical method forecasted.

### 3.5. Sensitivity Analysis

To evaluate the magnitude of the effects of the violations of assumptions presented in the previous section, we carried out a sensitivity analysis on our model by setting the observed attachment time to 1 day for all ticks that fell off on the first two days following the capture of their hosts. In this modified model, the means of the whole attachment times turned out to be 0.68–2.59 per cent shorter than those obtained in the original model (see Table 5). This result demonstrates that our model is not sensitive to the disturbances associated with the capture and transport of the host animals.

### 3.6. Seasonality of the Tick Infestation

Males were continuously present on the hosts throughout the year, with two smaller peaks of activity, one in the 4th week of April and a smaller one in late September.

Females had three active periods. The highest was in the 2nd week of June, and in the 4th weeks of August and September.

Nymphs dominated over the year with six peaks of activity: the 2nd (highest) and 4th weeks of April, two medium peaks were observed in the 2nd week of June and the 1st week of July and two smaller peaks in autumn in the 1st and 4th weeks of September.

It is usually considered that questing larvae have one big wave in July–August in the temperate climate of Europe. Our data confirmed this activity period, but seven other smaller waves were also detected. The engorgement of high number of larvae in autumn indicates that a lot of larvae run into the winter in this vulnerable form.

Three times the increased activity of all stages coincided, in the 4th week of April, the 2nd week of June, and the 4th week of September. For data see Figure 3.

If we consider the attachment times of ticks, they might have attached during the prior week as we counted the engorged ticks. However, this does not change the sizes and frequencies of peaks.

As the seasonal activity of ticks is concerned, from a mammal host a bimodal activity of each developmental stage of *I. ricinus* was described in Hungary [21,22], while in Sweden changes from unimodal to bimodal pattern from one year to the next were described from the same habitat and hosts [23]. Our weekly data show that the seasonal activities of all tick life stages consisted of several smaller or higher activity peaks throughout the year (Figure 3). The males showed two peaks, the females three, the nymphs 6 and the larvae 8; three times these peaks coincided. A four-year-long study of a tick-borne encephalitis focused on 250 km south-west from Budapest [24] revealed that the seasonal activities of the local *I. ricinus* populations changed from year to year, and seasonality diagrams were not similar to those typical of this urban habitat in Budapest. Although the main nymph activity in the first half of April and the peak of larval activity in late July were similar at the two sampling sites with a one-week delay. Activity periods could be roughly similar in various habitats, but the exact number and size of activity peaks seemed to differ by site from year to year, influenced by local microclimate and several biotic factors.

### 3.7. Juvenile Hedgehogs

Two hedgehogs were captured (both 0.44 kg) on 10 and 20 May: these must have been overwintered juveniles born in summer/autumn of the previous year. Six additional juveniles born in the study year were captured (three on 19–26 August, three between 8 and 16 October): two males and four females from litters of the two (spring and summer) gestation periods of hedgehogs. They had a lower tick burden than the adults (40.55 ticks/host to 91.45) and 7.69% of the females had an unsuccessful, not completed blood meal (5.57% on adult hosts). The intensity of infestation was also lower in the case of juveniles: 1.87 times more males, 1.64 times more females, 2.24 times more nymphs, and 2.73 times more larvae were found on adult hedgehogs than on juveniles. Only the proportion of males dropped off dead was very similar on juveniles (71.42%) and on adults (68.11%).

Among the captured hedgehogs, 10.25% were juveniles and 50% of them were found in October. This indicates that juveniles remain active the as long as the weather allows it, trying to accumulate energy resources for overwintering. The recorded tick data of the juvenile hedgehogs indicated that adult hedgehogs are more optimal hosts for ticks than juveniles (probably because of their body size), which mostly seems to be true for female ticks. However, statistical analyses did not demonstrate significant differences between the feeding success of ticks on adult and juvenile hedgehogs.

### 3.8. Additional Observations

In contrast to all other tick life stages, more than two-thirds of males (52) dropped off from hosts dead (68.42%). Only 24 males from the 76 were active, living, able to move after leaving the hosts. This indicates that most males seek females on the hosts till their energy supplies completely run out [25].

Not all females completed their blood meal successfully⁏ 22 from the 380 detached females (5.79%) were dead, or were smaller, dried, or shrunken. Females with an imperfect blood meal could not be statistically associated with the sex of the hosts, the season, host age, or the weather; imperfect blood meals were probably influenced by complex factors.

On 10 August, only once, a mating female/male pair fell down from their host, being active and healthy. This indicates that most mating happens and successfully finishes in the environment and on the hosts.

### 3.9. Effect of Weather

Most of the studied features of tick or hedgehog activities or the infestation showed no relationship with the recorded weather data (mean daily maximal and minimal temperatures, precipitation).

### 3.10. Zoonotic Potential of This Parasite/Host System

Human environments (parks, meadows, gardens) attract hedgehogs and the effective tick-carrying and spreading ability of this host, a successful urban species should be regarded as a risk factor for human public health, and their zoonotic role should be considered seriously [26,27]. Several zoonotic viral and bacterial agents were detected (mostly serologically) from them, such as Mycobacterium spp., *Salmonella* spp., *Coxiella* spp., *Mycoplasma* spp. [27], leptospirae [28], *Neoehrlichia mikurensis* [29], and viruses of canine distemper-morbilli group [30], rabies [31], and foot and mouth disease [32]. The main pathogens transmitted by ticks were also diagnosed in hedgehogs, the tick-borne encephalitis virus [33,34], *Borrelia* sl. [35] and *Anaplasma phagocytophilum* [36]. Helminths [37] and fleas [4] are also common parasites of hedgehogs.

We should also consider the zoonotic role of the ticks infesting urban hedgehogs, which could be infected by several zoonotic pathogens. A total of 31.5% of 2.417 urban ticks sampled from the vegetation in parks of Helsinki, Finland, carried at least one of the eleven pathogens surveyed in the study [37]. Similarly, ticks in an urban park were the sources of the first two human autochthonous tick-borne encephalitis (TBE) cases in Moscow, Russia [38].

Both the two main tick-borne zoonotic pathogens (tick-borne encephalitis virus and the Lyme disease) occur in Budapest. The TBEV is relatively rare, with 33 diagnosed cases in the last 20 years [39], while Lyme disease is more frequent. In the last 5 years the identified human Lyme cases varied between 1213 and 1640 in Hungary, and approximately 1/3 of these cases appeared in Budapest, and in its suburbs.

## 4. Conclusions

This study showed some details of engorgement of *Ixodes ricinus* on one of its natural host, the hedgehog. Statistical analysis determined the mean attachment times of various developmental stages of *I. ricinus*. Almost 70% of males dropped off the hosts dead. Seasonality of ticks consisted of several (4–6) activity peaks not only 1–2 times bigger, as was considered previously. On the first day after capture of the hosts, fewer females, nymphs, and larvae detached engorged than had been predicted, but this was not true for males.

As the hedgehog/tick pair seems to be an optimal host/parasite model to study mammal host/tick relations, in habitats with parallel dense hedgehog and tick populations (such as our study area), further studies could reveal novel details of this host–tick relationship.

## Figures and Tables

**Figure 1 microorganisms-11-00881-f001:**
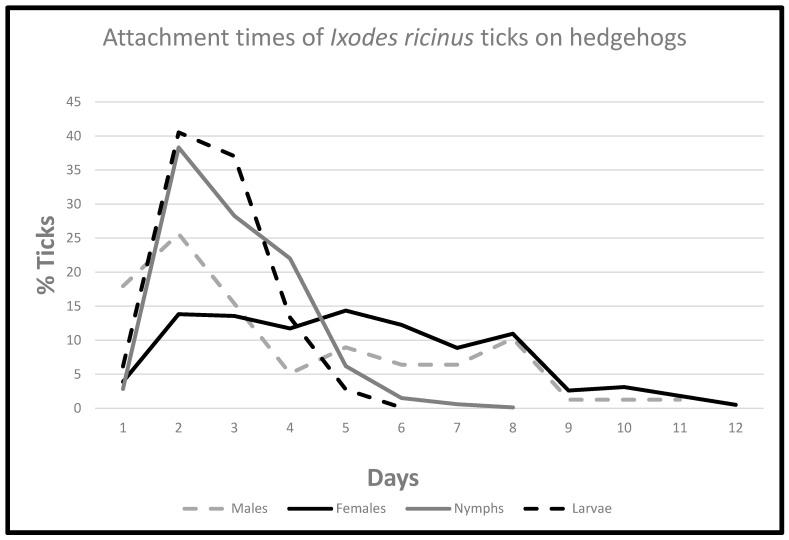
Attachment times of Ixodes ricinus ticks on hedgehogs.

**Figure 2 microorganisms-11-00881-f002:**
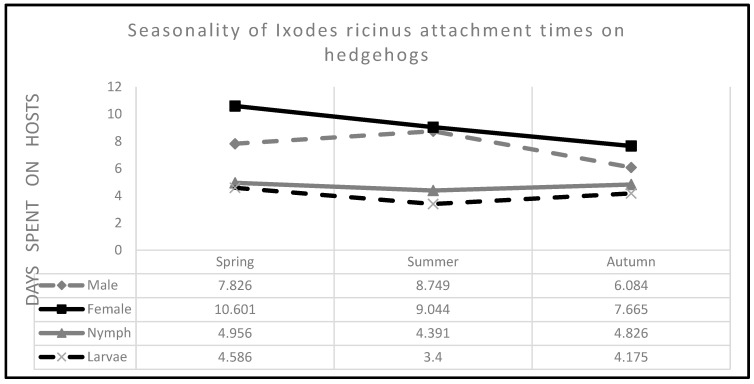
Seasonality of *Ixodes ricinus* attachment times on hedgehogs.

**Figure 3 microorganisms-11-00881-f003:**
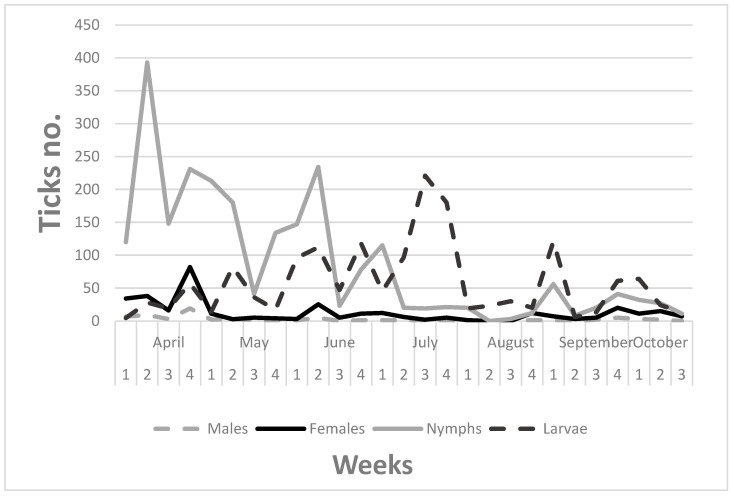
Seasonality of *Ixodes ricinus* ticks dropped off the hedgehogs.

**Table 1 microorganisms-11-00881-t001:** Seasonality data of captured hedgehogs and sampled ticks.

	Hedgehogs			*Ixodes ricinus*				
	M	F	All	M	F	N	L	All
Spring	7	11	18	49	213	1600	300	2162
Summer	11	16	27	13	95	804	1058	1970
Autumn	7	5	12	13	67	203	299	582
Total	25	32	57	75	375	2607	1657	4714
%	44	56	100	1.6	8	55.3	35.1	100

M—Males; F—Females; N—Nymphs; L—Larvae.

**Table 2 microorganisms-11-00881-t002:** Comparison of observed and modelled mean *Ixodes ricinus* numbers per hedgehog. The applied statistical model was a marginal generalised linear model fitting Poisson distribution.

Ticks		GEE Modelled Tick Number			Observed Mean Tick Number per Host		
		Tick No.	95% CI		Tick No.	SD	Tick Abundance
	Spring						
Males		2.78	1.26	6.14	2.78	3.44	50
Females		12.00	4.57	31.49	12.00	18.08	216
Nymphs		86.22	53.78	138.22	86.22	63.55	1552
Larvae		16.89	7.85	36.35	16.89	20.22	304
	Summer						
Males		0.52	0.21	1.29	0.52	0.89	14
Females		3.81	2.01	7.23	3.81	4.62	103
Nymphs		31.96	15.45	66.12	31.96	44.01	863
Larvae		43.85	25.41	75.66	43.85	45.31	1184
	Autumn						
Males		1.25	0.56	2.77	1.25	1.29	15
Females		5.08	2.80	9.22	5.08	3.92	61
Nymphs		12.33	6.93	21.94	12.33	9.20	148
Larvae		15.08	6.45	35.29	15.08	16.59	181

**Table 3 microorganisms-11-00881-t003:** Statistical comparison of seasonality of mean numbers of *Ixodes ricinus* per host between tick developmental stages. The applied statistical model was a marginal generalised linear model fitting Poisson distribution.

Seasons	Tick Stages	Estimate	Lower	Upper	*p* Values
Spring	Female/Male	4.32	3.21	5.82	<0.001
Spring	Larva/Male	6.08	2.54	14.55	<0.001
Spring	Nymph/Male	31.04	14.55	66.22	<0.001
S	Female/Larva	0.71	0.26	1.91	0.790
Spring	Nymph/Larva	5.11	2.28	11.45	<0.001
Spring	Female/nymph	0.14	0.05	0.35	<0.001

Summer	Female/Male	7.36	4.03	13.43	<0.001
Summer	Larva/Male	84.57	33.30	214.80	<0.001
Summer	Nymph/Male	61.64	32.90	115.49	<0.001
Summer	Female/Larva	0.09	0.04	0.18	<0.001
Summer	Nymph/Larva	0.73	0.35	1.52	0.680
Summer	Female/nymph	0.12	0.06	0.23	<0.001

Autumn	Female/Male	4.07	2.35	7.03	<0.001
Autumn	Larva/Male	12.07	9.22	15.79	<0.001
Autumn	Nymph/Male	9.87	5.81	16.75	<0.001
Autumn	Female/Larva	0.34	0.19	0.61	<0.001
Autumn	Nymph/Larva	0.82	0.43	1.55	0.830
Autumn	Female/nymph	0.41	0.30	0.57	<0.001
**Tick stages**					
Males	Summer/Spring	0.19	0.07	0.51	<0.001
Males	Autumn/Spring	0.45	0.18	1.15	0.115
Males	Autumn/Summer	2.41	0.88	6.64	0.104

Females	Summer/Spring	0.32	0.12	0.84	0.015
Females	Autumn/Spring	0.42	0.16	1.09	0.085
Females	Autumn/Summer	1.33	0.64	2.76	0.625

Nymphs	Summer/Spring	0.37	0.18	0.77	0.004
Nymphs	Autumn/Spring	0.14	0.08	0.27	<0.001
Nymphs	Autumn/Summer	0.39	0.18	0.84	0.011

Larvae	Summer/Spring	2.60	1.18	5.71	0.013
Larvae	Autumn/Spring	0.89	0.34	2.33	0.959
Larvae	Autumn/Summer	0.34	0.15	0.80	0.009

No significant relationship was found between the intensity of tick infestation and the body weight and sex of the hedgehogs.

**Table 4 microorganisms-11-00881-t004:** Times spent on hosts by various *Ixodes ricinus* developmental stages (in days).

	Detached Engorged Ticks (Days)														
	1	2	3	4	5	6	7	8	9	10	11	12	13	14	15
Males	14	21	12	4	7	6	5	7	1	1	1				
Females	17 *	55	47	46	52	47	41	31	15	18	2	2	4	2	1
Nymphs	64 *	913	726	618	171	42	18	8	3						
Larvae	105 *	732	561	217	48	6									

*—Significantly lower number than value of the second day. To compare the abundances, we applied negative binomial generalised linear mixed model.

**Table 5 microorganisms-11-00881-t005:** Modelled and experienced attachment times of *Ixodes ricinus*.

Ticks	Season	Modelled Attachment Times			Experienced Attachment Times		
		Days	95% CI		Days	SD	Tick Abundance
	Spring						

Males		7.826	6.486	9.442	3.42	2.625	50
Females		10.601	9.591	11.718	5.19	2.943	216
Nymphs		4.956	4.567	5.378	2.512	1.257	1552
Larvae		4.586	3.977	5.263	2.717	1.059	304
	Summer						
Males		8.749	7.005	10.926	3.071	3.131	14
Females		9.044	8.281	9.877	4.267	2.72	103
Nymphs		4.391	4.190	4.601	2.638	0.941	1184
Larvae		3.40	3.207	3.605	1.933	0.784	863
	Autumn						
Males		6.084	5.17	7.16	3.10	1.882	15
Females		7.665	6.90	8.515	3.992	2.18	61
Nymphs		4.826	4.406	5.286	2.75	1.183	148
Larvae		4.175	3.855	4.522	2.467	0.924	181

## Data Availability

The datasets generated during the current study are available from the corresponding author on reasonable request.

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
