# Peer review of "Features of Engorgement of Ixodes ricinus Ticks Infesting the Northern White-Breasted Hedgehog in an Urban Park"

_microorganisms, 2023, doi:10.3390/microorganisms11040881_

Round 1

Reviewer 1 Report

The study contributes to the knowledge of tick-hedgehog associations on the example of an isolated population of Erinaceus roumanicus living in an urban park located on a Danube island in Budapest. The authors developed an original approach to assess engorgement times of the ticks attached to hedgehogs and seasonal activities of tick larvae, nymphs and adults. The achieved results are interesting but the manuscript in its form is not ready for acceptance and needs substantial revisions.

First of all, thorough proof-reading of the text with the help of a native speaker, expert in the field, should be carried out.

 A few specific comments:

The abstract should contain and introductory and a concluding sentence.

Were the captured hedgehogs marked so as to record recaptures? Were recaptures noticed?

How were the collected ticks stored and identified?

More specific comments and a few corrections are included in the attached file

Author Response

Answer to Rev 1 and admin.

We thank Your time effert and help in improving this manuscript. Please find below our answers to Your questions.

Line 76: - The ticks were stored at 4C for 1-2 days before species identification. immediatley after itt he ticks were discarded.

line 175 -185: Of course this Table is not correct that way. In my computer either as word or Pdf file it is correct. I do not know what could happen to this Table during transforming it to Your submission system. Hopefully during this resubmission ot going to be good.

line 223. I do not understand here the question about „which data”. For us it is clear.

Table 2A. Proportions of the developmental stages according to seasons.

Table 2B. comparison of the same developmental stage in various seasons.

line 239. The same as with Table 1. Here in my computer the format of the Table 4 is correct. I do not know what could happen during its submission. Of course I did not upload these Tables into Your submission system that way.

line 288. I do not know, what could happen here. In my computer here we have an other (table 3) which is missing from the submitted form of the manuscript. I submitted the manuscript for 3-4 days I suffered a lot with it. Some kind of mistakes could have happened, I made a mistake dring the submission process, or what happened, I do not know. In the resubmitted version I hope everything going to be OK.

line 295. No literature data. But here we found a phenomenon, which has never been mentioned before in the literature. Such observations take science forward. We had to give some explanations, and we think our version is a possible answer. The word speculation for me means that the speculated thing is not true. Observing a novel phenomenon in tick-host relation is useful and not speculation.

line 367. The sentence was omitted.

Reviewer 2 Report

The manuscript is not within the scope of the scientific journal Microorganisms. It has serious writing problems and lacks solidity in the presentation of data and information. Reformulate and submit to a journal where the manuscript fits, such as Parasitologia (ISSN 2673-6772).

- Why is the abstract divided into paragraphs? The Abstract needs to be written in a single paragraph.

- The abstract has no background and does not describe the objectives of the study.

- Line 16: Replace "I. ricinus" with "Ixodes ricinus".

- Overall, the abstract is pretty bad. It appears that random excerpts from the study were copied and pasted without any connection. The abstract must contain background, objectives, methods, main results and conclusions.

- Line 38: Revise "Their habitats ae mainly forests".

- Line 43: Replace "I. hexagonus" with "Ixodes hexagonus", because it is the first citation of this taxon.

- Line 42-45: This paragraph is out of context. The study is about Ixodes ricinus and the introduction does not provide substantial data about the problem that is the Ixodes ricinus tick. In addition to this lack, these lines address aspects of I. hexagonus, which are not the problem in question. The introduction needs to be better supported.

- Line 52-53: Revise "on the Danube, i the centre of Budapest".

- The methodology is not clear. Some questions: Was the animals already parasitized by ticks? Were they infested in the lab? How was the collection and observation of ticks? How were the ticks identified as Ixodes ricinus? It is not possible to understand the methods as they are described.

- Line 94: After the sentence "The conditions of applicability of these methods are as follows." change the period to a colon.

- Line 101: Replace "We fitted Weibull distribution" with "Weibull distribution was adapted".

- Overall, the text needs language revision.

- Tables are messy, misaligned, and captions are not self-explanatory.

- The title is about Ixodes ricinus, the objectives too, however the results are more diverse and present another species. The manuscript does not appear to be ready. It looks like a primary draft was made and submitted to a magazine. I strongly suggest authors rewrite it, better understanding the idea they are studying.

Author Response

We tried to rewrite the paper, but as we did not get too many exact, concrete criticism, we had to imagine Your taste in general.

Reviewer 3 Report

The topic of the manuscript “Features of engorgement of Ixodes ricinus ticks infesting the northern white-breasted hedgehog in an urban park” could be  interesting for scientific community and would deserve publication but I am not sure that it is appropriated for the journal scope because neither hedgehogs nor ticks are microorganisms. The final decision would be taken by the editor.

Nevertheless, the manuscript needs to be improved in the following:

The table formats should be homogenized throughout the entire manuscript. In my opinion the table 2 design is the best because in tables 1 and 4 the content is not correctly aligned in columns and it is hard to understand.

The results and discussion are presented together but I consider that in general, results are presented with very little discussion. The biological significance  of the findings such as the prevalence of each tick stage, the variability of attachment time according seasons or tick stage should be deeper analyzed and their implications for tick population.

In the same way, the exact correspondence between the modeling  and the observed of tick numbers presented on the table 3 should be also discussed.

What is the biological significance of results presented on the table 4?

Taking into account that the density of hedgehogs and Ixodes ricinus ticks are similar in these special conditions on an island situated on the Danube with an urban environment which is not the same  in natural conditions which are the implications of the findings comparing this situation with that on the forests? It would be reflected in the conclusions.

Finally, some English improvements and typing errors should be addressed. For example, lines 38, 53, 92 have typing errors. Sentence in lines 59-60 should be improved maybe dividing it in two sentences. It is not correct beginning a sentence with a number (line 172). The sentence redaction in paragraph on lines 187-193 should be improved in order to express the ideas more clearly.

Author Response

Thank You for Your work, time and help in improving this manuscript. Please find our answers to Your questions:

The table formats should be homogenized throughout the entire manuscript. In my opinion the table 2 design is the best because in tables 1 and 4 the content is not correctly aligned in columns and it is hard to understand. See below. Table 1 and 4 somehow went wrong during submission. We did not sent tables in such form. It was not hard to understand, but it was completely a mass and ununderstable. Hopefully during submission of the revised version Tables remain in their original form.

The results and discussion are presented together but I consider that in general, results are presented with very little discussion. The biological significance  of the findings such as the prevalence of each tick stage, the variability of attachment time according seasons or tick stage should be deeper analyzed and their implications for tick population.

This is a descriptive work. We provide lot of numbers and data of a tick-host relation of a certain, special environment. We should know, that such features of the tick-hedgehog relation exists. Implication of our data to other species or habitats would be a mistake. e.g. We could find males on the host for 11 days (Table 4). What to discuss more deeper? This was the situation here. In other fields, forests etc. slightly different similar data could be found. I think we gave the concluding sentences, but basically it is description of a relation, it is quite evident and we can not discuss these data more deeply as these are facts here. As I have learnt from my past, it is best to avoid discussing, concluding sentences, because the author always get critics for these sentences from the reviewers and only a never ending debate starts.

In the same way, the exact correspondence between the modeling  and the observed of tick numbers presented on the table 3 should be also discussed.

It is much more mathmatics and statistics than parasitology. Most parasitologists do not understand the statistical background, statisticians do not understand the biological background. A statistician/mathematician made a mathematical model to predict the engorgement times of various life stages of ticks. (written detailed in the Appendix). We had to do that, as we could not know when ticks attached to the host. In most places the model worked and predicted well the observed data. Table 3. What to discuss more and deeper. I think it won’t be more clear, but more confuse.  

What is the biological significance of results presented on the table 4?

It shows how long we could find detached tick individuals (and how many) after the day of capture (day 0). During the submission process, TAble 1 and 4 were completely mixed. In the sent original version both Tables were intact. As in the new, revised version. We hope after submitting the tables remains  in their original form. We did not send tables as such a mass, as Table 1 and 4 appeared in the final version. Hopefully in its original ordered form, they will be evident, and easier to understand.

Taking into account that the density of hedgehogs and Ixodes ricinus ticks are similar in these special conditions on an island situated on the Danube with an urban environment which is not the same  in natural conditions which are the implications of the findings comparing this situation with that on the forests? It would be reflected in the conclusions.

It was not written anywhere in this paper, that features of this tick-parasite relation could be implicated to any other biotops. It is a special urban park habitat. All meadows, gardens, forests with their other type of tick and hedgehog (host) populations are, and must be different. 

Finally, some English improvements and typing errors should be addressed. For example, lines 38, 53, 92 have typing errors. Done.

Sentence in lines 59-60 should be improved maybe dividing it in two sentences. Done.

It is not correct beginning a sentence with a number (line 172). The sentence redaction in paragraph on lines 187-193 should be improved in order to express the ideas more clearly. Done (tried).

Round 2

Reviewer 1 Report

The manuscript has been improved, but there are still some inconsistencies that should be fixed. The language should be improved further. 

1/ Last paragraph of introduction, L. 126 – results of seasonality of questing ticks are not included in the results, please revise the aims of the study accordingly.

2/ Results, L. 270  – It is written that “A total of 4,745 ticks ….. and further that “99.35% were I. ricinus”

However, in Table 1 the total number 4,745 is given for I. ricinus. Please revise the heading of the table and also the remaining text including the title of the paper in case all ticks were included in the analyses, not only I. ricinus.

3/ P. 5, L. 294-305. Please avoid repetition in the two paragraphs.

4/ Abundance of ticks, L. 309. Please delete “questing”    

5/ Time od Engorgement, L. 392. Please make clear that 12 days is the maximum time estimated for engorgement of females.

6/ Table 4. Please add explanation to 1, 2, 3…. (i.e. days)

7/ L. 465-469. The authors have right to express their opinion, but because they have no proofs, even not from literature, I suggest they write “As the capture of the hosts is a stress factor, we hypothesize that the temporarily elevated stress hormones (glucocorticoids) could have caused such an effect”

8/ Table 5, right column, should be “Tick abundance”

9/ Figure 3, please italicize Ixodes ricinus in the caption and correct the legend in the figure, currently “Weeks” covers “Nymphs”

10/ L. 546-548 Please correct to “Activity periods could be roughly similar in various habitats, but the exact number and size of activity peaks seem to differ by site from year to year, influenced by local microclimate and several biotic factors.”

11/ Please check the list of references once again, Latin names should be italicized, but currently some are and some are not.

Author Response

Reviewer 1.

Thank You for Your help again in improving this manuscript . We tried to do our best to follow the instructions. During submitting, the review, and handling by the editor’s office some changes were generated (italics, tables, figures, captions, line numbers were changed) etc. which originally were not included in the text. Hopefully outlook of this final version remains fixed.

The manuscript has been improved, but there are still some inconsistencies that should be fixed. The language should be improved further. 

1/ Last paragraph of introduction, L. 126 – results of seasonality of questing ticks are not included in the results, please revise the aims of the study accordingly.

Table 3 and Figure 3 is about seasonality of questing ticks. We intended to indicate that in this final summerizing sentence of the introduction. (which sentence was required by one of the reviewers. I change everything but I do not understand what to rewrite or omit.

2/ Results, L. 270  – It is written that “A total of 4,745 ticks ….. and further that “99.35% were I. ricinus”

However, in Table 1 the total number 4,745 is given for I. ricinus. Please revise the heading of the table and also the remaining text including the title of the paper in case all ticks were included in the analyses, not only I. ricinus.

Table 1 was changed as requested. The difference is so low, we missed the numbers. I. ricinus in the title was introduced by suggestion of an other reviewer. I rewrite but how to omit and include the same words in the same time in the same title? Finally I omitted.

3/ P. 5, L. 294-305. Please avoid repetition in the two paragraphs. Page and line numbers cause more problems than they help. Line numbers in the manuscript I sent in, line numbers the reviwer is writing about, and line numbers I see in the version I got back from the journal are different. I can not localize the site of the above repetition. I would like to ask editor or the journal to send me the version the reviewer saw.

4/ Abundance of ticks, L. 309. Please delete “questing” If You meant the word “questing” from the first line of chapter 3.2 , I did.

5/ Time od Engorgement, L. 392. Please make clear that 12 days is the maximum time estimated for engorgement of females. The sentence was rewritten. Female predicted attachment time was 10,6 days (Table 5) which was a good estimate as 97,1% of females detached the host within this time-span.

6/ Table 4. Please add explanation to 1, 2, 3…. (i.e. days) Table 4 headings were rewritten.

7/ L. 465-469. The authors have right to express their opinion, but because they have no proofs, even not from literature, I suggest they write “As the capture of the hosts is a stress factor, we hypothesize that the temporarily elevated stress hormones (glucocorticoids) could have caused such an effect”

Thank You, the sentence was included in the text. (311-312). We think here with this first day decrease of detached ticks we found something novel.

8/ Table 5, right column, should be “Tick abundance” Done.

9/ Figure 3, please italicize Ixodes ricinus in the caption and correct the legend in the figure, currently “Weeks” covers “Nymphs” yes, done.

10/ L. 546-548 Please correct to “Activity periods could be roughly similar in various habitats, but the exact number and size of activity peaks seem to differ by site from year to year, influenced by local microclimate and several biotic factors.” Done.

11/ Please check the list of references once again, Latin names should be italicized, but currently some are and some are not. I checked, all latin names seem to be italicized.

Reviewer 2 Report

Dear authors,

First, the study can (if improved) and obviously in the humble opinion of this reviewer, be published in another journal, as it does not understand the scope of the "Microorganism" journal, even in the proposed "Special Issue", as the study does not correlate with tick-borne pathogens.

Second, I'm sorry if you missed the numerous comments I made in the previous review. I've put them back here (if you like). I didn't make them for "taste", but as a way to, in fact, improve the study. Thus, they follow, again, for an adequate response with corrections to each item (if you wish to comply):

The manuscript is not within the scope of the scientific journal Microorganisms. It has serious writing problems and lacks solidity in the presentation of data and information. Reformulate and submit to a journal where the manuscript fits, such as Parasitologia (ISSN 2673-6772).

- Why is the abstract divided into paragraphs? The Abstract needs to be written in a single paragraph.

- The abstract has no background and does not describe the objectives of the study.

- Line 16: Replace "I. ricinus" with "Ixodes ricinus".

- Overall, the abstract is pretty bad. It appears that random excerpts from the study were copied and pasted without any connection. The abstract must contain background, objectives, methods, main results and conclusions.

- Line 38: Revise "Their habitats ae mainly forests".

- Line 43: Replace "I. hexagonus" with "Ixodes hexagonus", because it is the first citation of this taxon.

- Line 42-45: This paragraph is out of context. The study is about Ixodes ricinus and the introduction does not provide substantial data about the problem that is the Ixodes ricinus tick. In addition to this lack, these lines address aspects of I. hexagonus, which are not the problem in question. The introduction needs to be better supported.

- Line 52-53: Revise "on the Danube, i the centre of Budapest".

- The methodology is not clear. Some questions: Was the animals already parasitized by ticks? Were they infested in the lab? How was the collection and observation of ticks? How were the ticks identified as Ixodes ricinus? It is not possible to understand the methods as they are described.

- Line 94: After the sentence "The conditions of applicability of these methods are as follows." change the period to a colon.

- Line 101: Replace "We fitted Weibull distribution" with "Weibull distribution was adapted".

- Overall, the text needs language revision.

- Tables are messy, misaligned, and captions are not self-explanatory.

- The title is about Ixodes ricinus, the objectives too, however the results are more diverse and present another species. The manuscript does not appear to be ready. It looks like a primary draft was made and submitted to a magazine. I strongly suggest authors rewrite it, better understanding the idea they are studying.

Respectfully,

Author Response

The manuscript is not within the scope of the scientific journal Microorganisms. It has serious writing problems and lacks solidity in the presentation of data and information. Reformulate and submit to a journal where the manuscript fits, such as Parasitologia (ISSN 2673-6772). Editors will decide about where this manuscript fits.

- Why is the abstract divided into paragraphs? The Abstract needs to be written in a single paragraph. The abstract was not written in paragraphs. During submitting (days) several editing problems appeared in the paper e.g. changes and lack of Figures. Similarly line numbers are differ between the versions I sent in, and I got back.

- The abstract has no background and does not describe the objectives of the study. The study has no background. The objective is obviously is to study the tick-host relation in a special environment with abundant tick-host populations. As I see, such empty general sentences are best to avoid, but if the reviewer needs it, I included.

- Line 16: Replace "I. ricinus" with "Ixodes ricinus". Done.

- Overall, the abstract is pretty bad. It appears that random excerpts from the study were copied and pasted without any connection. The abstract must contain background, objectives, methods, main results and conclusions. Thanks for explanining how an abstract should look like. This work has no background, and has no methods, counting ticks are not a method. The applied statistics can not be described in the Abstract. Results are complex and shown in several figures and Tables. I can not summerized them in 3 sentences. With some sentences I tried to make the Abstract more readable.

- Line 38: Revise "Their habitats ae mainly forests". Done.

- Line 43: Replace "I. hexagonus" with "Ixodes hexagonus", because it is the first citation of this taxon. Done.

- Line 42-45: This paragraph is out of context. The study is about Ixodes ricinus and the introduction does not provide substantial data about the problem that is the Ixodes ricinus tick. I do not understand this sentence. Probably You also not. In addition to this lack, these lines address aspects of I. hexagonus, which are not the problem in question. We did not intended to make this manuscript more complex by analysing data of I. hexagonus. The very small number of I. hexagonus did not allowed us statistical analysis, and without statistics, there was no need to write about this species too much.  The introduction needs to be better supported. ??????

- Line 52-53: Revise "on the Danube, i the centre of Budapest". Done.

- The methodology is not clear. Some questions: Was the animals already parasitized by ticks? Were they infested in the lab? If You could not understand this, You did not understand anything from this manuscript. How could I collect a hedgehog from natural environment without ticks? How could I guarantee, that a hedgehog is free of ticks? Why should I put ticks on the host in the lab? How was the collection and observation of ticks? As I see it is written clearly. It s a very simple experiment concerning the field-work part with quite difficult/complex statistical analyses. Daily we collected the detached ticks from a tray put below the cages of the hedgehogs. It was written. What other details are necessary? How were the ticks identified as Ixodes ricinus? A collegue from the parasitology department of the Univ. of  Veterinary Science Budapest identified them by light microscope, ont he base of taxonomy criteria I do not know, and I do not feel it worth giving here in detail. It is not possible to understand the methods as they are described. I read again carefully this part, for me it is clear. This mat and meth. part of this work is very simple and evident, I really can not understand what is not possible to understand here. If You could not understand the collection of the hosts, counting their ticks, than You could not understand the whole paper, probably that is why You gave a negative opinion about it.

- Line 94: After the sentence "The conditions of applicability of these methods are as follows." change the period to a colon. Done.

- Line 101: Replace "We fitted Weibull distribution" with "Weibull distribution was adapted". Done.

- Overall, the text needs language revision.

- Tables are messy, misaligned, and captions are not self-explanatory.

- The title is about Ixodes ricinus, the objectives too, however the results are more diverse and present another species. ??????? The manuscript does not appear to be ready. It looks like a primary draft was made and submitted to a magazine. I strongly suggest authors rewrite it, better understanding the idea they are studying.